# Biomimetic Superhydrophobic Films with an Extremely Low Roll-Off Angle Modified by F_16_CuPc via Two-Step Fabrication

**DOI:** 10.3390/nano12060953

**Published:** 2022-03-14

**Authors:** Pengchao Zhou, Tengda Hu, Yachen Xu, Xiang Li, Wei Shi, Yang Lin, Tao Xu, Bin Wei

**Affiliations:** 1School of Mechatronic Engineering and Automation, Shanghai University, Shanghai 200444, China; pczhou@shu.edu.cn (P.Z.); hutengda@shu.edu.cn (T.H.); xuyachen1985311@163.com (Y.X.); shiwei@shu.edu.cn (W.S.); xtld@shu.edu.cn (T.X.); 2School of Materials Science and Engineering, Shanghai University, Shanghai 200444, China; 2575154029@shu.edu.cn

**Keywords:** superhydrophobic surfaces, molecular sieves, polydimethylsiloxane, F_16_CuPc, water contact angle

## Abstract

Superhydrophobicity is the phenomenon of which the water contact angle (WCA) of droplets on a solid surface is greater than 150°. In the present paper, we prepare a superhydrophobic film with a structure similar to the surface of a lotus leaf, which is composed of polydimethylsiloxane (PDMS), zinc oxide (ZnO), a molecular sieve (MS) and 1,2,3,4,8,9,10,11,15,16,17,18,22,23,24,25-hexadecafluorophthalocyanine copper(II) (F_16_CuPc). The F_16_CuPc was used as the modifier to reduce the surface energy of the biomimetic micro-nanostructure. With the introduction of F_16_CuPc, the superhydrophobic properties of the surface were enhanced so that the WCA and water roll-off angle could reach 167.1° and 0.5°, respectively. Scanning electron microscopy, X-ray energy spectrometry, and X-ray photoelectron spectroscopy analyses verified that the enhanced superhydrophobic properties of the film were mainly attributed to the modification of F_16_CuPc. Finally, thermal, mechanical, and chemical stability studies, as well as the influences of UV and underwater immersion on the superhydrophobic film were investigated. This developed two-step fabrication method may be a potential direction for superhydrophobic surface fabrication due to its simple process, excellent superhydrophobic property, and favorable stability.

## 1. Introduction

The superhydrophobic phenomenon in which the water contact angle (WCA) of droplets on a solid surface is larger than 150°, is of great significance for practical applications, such as microfluidics, self-cleaning coatings, oil/water separation, drag reduction, and antifouling [1,2,3,4,5,6]. Although a large number of endeavors have been attempted and the plentiful superhydrophobic materials have been investigated, it is still difficult to meet the demands for superhydrophobic surface functionality in practical applications due to the high cost or complicated fabrication process [7,8,9,10].

To fabricate superhydrophobic surfaces, both the rough morphology and the hydrophobic composition on the surface are indispensable, since the wettability of a surface is mainly governed by the combination of the surface geometrical structure and the chemical composition [11,12,13,14,15]. The molecular sieves (MSs), with a carved hexahedron the size on the micron level [16,17,18], have been widely investigated as catalysts, gas separation, and adsorbents. This micro structure is suitable to construct the rough substrate for the superhydrophobic surface. Therefore, using the MS as the substrate to construct the rough morphology of the superhydrophobic surfaces may be an interesting attempt [19,20]. Considering the relatively poor hydrophobicity of the MS, a suitable modifier needs to be incorporated to reduce the surface energy and to achieve the superhydrophobic property of the MS surface. To fabricate the superhydrophobic surface, polydimethylsiloxane (PDMS) was also selected as the matrix, due to its low surface energy, favorable stability, and hydrophobic properties. In addition, zinc oxide (ZnO) nanoparticles were employed to serve as the modifier to increase the roughness of the surface, and finally to achieve the superhydrophobic substrate. For example, Chakradhar et al. prepared superhydrophobic coatings based on the ZnO-PDMS nanocomposite using the wet chemical method [21]. The ZnO nanoparticles on the surface were linked to form a hierarchical network with an average diameter of 100–150 nm. The WCA increased from 108° to 155° with a roll-off angle less than 5°. Wang et al. prepared ZnO-PDMS superhydrophobic coatings by the electrodeposition-grafting modification method, which had a superhydrophobic property and produced an excellent photocatalytic performance [22]. However, most methods for constructing superhydrophobic surfaces include plasma etching, sol–gel method, and self-assembly techniques, which were tedious, complicated, and high cost [23,24,25,26,27]. Therefore, it is necessary to develop a simple, economic, and environmentally friendly fabrication method to obtain the stable superhydrophobic surface.

In this study, the mixed solution of MS, ZnO, and PDMS was spin-coated and a stacked substrate was constructed. This stacked substrate presented a two-level biomimetic lotus-leaf structure and exhibited excellent superhydrophobic properties. More importantly, the 1,2,3,4,8,9,10,11,15,16,17,18,22,23,24,25-hexadecafluorophthalocyanine copper(II) (F_16_CuPc) was firstly used as a modifier to improve the performance of the superhydrophobic surface based on the stacked substrate. Since the introduction of F_16_CuPc enhanced the superhydrophobic property of the stacked substrate, the WCA and the water roll-off angle (WRA) reached 167.1° and 0.5°, respectively. The optimized WCA and WRA demonstrated the superiority of the F_16_CuPc serving as the modifier to improve the superhydrophobic properties of the stacked substrate. On the other hand, the stability of the biomimetic superhydrophobic films with the modification of F_16_CuPc is also investigated. The results demonstrate that the modified biomimetic superhydrophobic films have excellent physical and chemical stability. This simple fabrication process can be widely applied in ceramics, glass, metal, and other surfaces. 

## 2. Experimental Section

### 2.1. Materials

The ZnO nanoparticle was purchased from Hengqiu Co., Ltd. (Suzhou, China) and SYLGARD 184 SILICON ELASTOMER (PDMS and curing agent) was received from Shanghai Deji Trading Co., Ltd. (Shanghai, China) (Dow Chemical (China) Investment Co., LTD (Shanghai, China)). F_16_CuPc was purchased from Sigma Co., Ltd., (Tokyo, Japan) and the MS were synthesized by our group (Appendix A). Cyclohexane (99.8 wt.%), absolute ethanol (99.5 wt.%), aluminum isopropoxide (97 wt.%), sodium hydroxide (NaOH), tetramethylammonium hydroxide (TMAOH, 25 wt.%), calcium chloride (CaCl_2_), and ethylene glycol (EG, 99.5 wt.%) were purchased from Sinopharm Chemical Reagent Co., Ltd. (Shanghai, China). All materials were used without any further purification. 

### 2.2. Preparation of the Films

Firstly, the mixed solution of MS, ZnO, and cyclohexane was prepared in advance with the optimal ratio of 1:1:8 and stirred for 2 h with a cleaned magnetic stirrer. PDMS and curing agent were mixed in the mass ratio of 10:1, and then stirred for 5 min to obtain the fully mixed PDMS solution. Next, the mixed PDMS solution was degassed in a vacuum box for 20 min to expel the air bubbles. Subsequently, the degassed PDMS was spun on a clean glass substrate at 3000 rpm for 30 s, and then annealed for 5 min to obtain a semi-solidified PDMS film. After that, the mixed solution of MS, ZnO, and PDMS was spin-coated on the semi-solidified films with a speed of 3000 rpm for 30 s. The sample was annealed at 100 °C for 3 h to remove the cyclohexane solution. At this stage, the stacked substrate with a structure of PDMS/PDMS:MS:ZnO was obtained. Finally, the stacked substrate was transferred to a high vacuum chamber and the F_16_CuPc of 50 nm thickness was thermally deposited on the stacked substrate under the base pressure of 1 × 10^−4^ Pa. Figure 1 illustrates the schematic diagram of the biomimetic superhydrophobic film.

### 2.3. Characterization

The temperature experiment was conducted at room temperature, where the samples were first annealed at different temperatures and then cooled to room temperature for the subsequent characterization. The wear tests were performed by applying 500 N and 1000 N weights on 1000 # sandpaper coated on the superhydrophobic films, and then the sandpaper was pulled with a constant speed of 1cm/s for 10 cm. The tape stripping experiments were performed by completely laminating the sample surface with the tape and then peeling it off. 

The morphology of the film was captured by the scanning electron microscopy (SEM) (Hitachi S-4800 and Zeiss Gemini 300). Energy-dispersive X-ray spectroscopy (EDS) and X-ray photoelectron spectroscopy (XPS) were used to conduct the elemental analysis of the films. The WCA was measured by an SL200KS optical contact angle measuring instrument and the interfacial tension measuring instrument. The WRA was measured by a Dataphysics OCA20 contact angle measuring instrument. The image of the water droplets rebound process on the solid surface was captured by the Optronis CP70-2-M/C-1000 high-speed camera. 

## 3. Results and Discussion

The stacked substrate exhibited excellent superhydrophobic properties with a static WCA of 162.3° and a WRA of 3.2°, which was better than most of the superhydrophobic surfaces that were composed of ZnO and PDMS [21,28,29]. To further improve the WCA and the WRA of the substrate, the additional modifier with a low surface energy needed to be introduced. F_16_CuPc is widely used in organic electronic devices as the buffer layer between the active layer and metal electrodes, due to its superior electrical and optical properties [30,31,32,33,34,35]. However, the wettability of F_16_CuPc with low surface energy is little utilized. To assess the feasibility of using F_16_CuPc as a modifier to improve the superhydrophobic properties of the stacked substrate, the WCA and the surface energy of the pure F_16_CuPc film deposited on glass were measured, as shown in the Appendix A. As shown in Appendix A, the water and EG contact angles were 105.6° and 77.9°, respectively. The surface energy of the pure F_16_CuPc film was calculated as 25.97 mN/m, based on the Owens–Wendt–Kaelble equation as shown in Equations (1) and (2) [36]. The surface energies were calculated based on the WCA and EG contact angle (EGCA).
(1)$$1+\;\cos\theta =2\cdot (\sqrt{\gamma^{d}\gamma_{L}^{d}}+\sqrt{\gamma^{h}\gamma_{L}^{h}}\;$$
(2)$$\gamma \;=\gamma^{d}+\gamma^{h}$$
where *θ* denotes the value of the contact angle measured for different droplets, $\gamma_{L}$, $\gamma_{L}^{d}$, and $\gamma_{L}^{h}$ denote the surface energy, dispersion force component, and polar force component of different droplets, respectively. γ, $\gamma^{d}$, and $\gamma^{h}$ denote the surface energy, dispersion force component, and polar force component of the measured film, respectively. The roughness of the pure F_16_CuPc film deposited on the glass was measured by AFM, since morphology plays a crucial role in determining the wettability of the film. As shown in Appendix A, the F_16_CuPc film exhibited a consecutive morphology with the root-mean-square roughness (RMS) of 2.05 nm and the maximum peak-to-valley height of 13.46 nm. The consecutive and smooth morphology, the high WCA, and low surface energy of the F_16_CuPc demonstrated that it may be an alternative modifier to further improve the superhydrophobic properties of the stacked substrate. Therefore, the F_16_CuPc film with a 50 nm thickness was thermally deposited on the stacked substrate to enhance the superhydrophobicity of the surface.

Figure 2 shows the SEM images of the F_16_CuPc-modified superhydrophobic film. The two-level structures were observed after the F_16_CuPc modification. In this micro-nanostructure, the MS particles served as the primary structure on a micron level, while ZnO nanoparticles and the F_16_CuPc served as the secondary structure on a nanometer level. This biomimetic lotus-leaf surface structure provided sufficient roughness for the superhydrophobic surface [37,38]. 

To further investigate the distribution of F_16_CuPc on the stacked substrate, EDS analyses were conducted. As shown in Figure 3, Al, Zn, and F elements present the distribution of MS, ZnO, and F_16_CuPc, respectively. The F element presented a relatively symmetrical distribution on the stacked substrate surface, and, to a certain extent, gathered on the surface of MS, which may be attributed to the adsorption of micro-holes on the surface of MS [39,40]. The subsequent XPS measurement in Appendix A further verifies the existence of F_16_CuPc on the surface of the stacked substrate. This homogeneously distributed F_16_CuPc on the surface could further reduce the surface energy and finally improve the superhydrophobic properties of the stacked substrates.

Figure 4a–f show the WCA, EGCA, and the WRA of the stacked substrate and the F_16_CuPc-modified biomimetic superhydrophobic film, respectively. It can be clearly observed that the F_16_CuPc-modified biomimetic superhydrophobic film showed better superhydrophobic properties with a large WCA of 167.1° and a small WRA of 0.5° than that of the stacked substrate with a WCA of 162.3° and a WRA of 3.2°. Figure 4g shows an image of a side water jet incident on the surface of the modified substrate. The outgoing angle of the water jet (30°) was slightly smaller than the incident angle (35°), which demonstrated that the energy loss in the whole water jet reflection process was small. This confirmed the excellent superhydrophobic properties of the modified surface [39]. The self-cleaning ability of the F_16_CuPc-modified biomimetic superhydrophobic film was investigated. As shown in Figure 4h,i, the dust particles are dispersed on the surface of the glass and the tilted angle of the superhydrophobic film is 30°. After the continuously dropping of deionized water droplets on both surfaces, we found that most of the dust particles and droplets remained on the glass substrate, while the dust particles on the surface of the superhydrophobic film could completely fall off with the water droplets. This verified that the superhydrophobic films have an self-cleaning excellent capability.

Figure 5 represents the image of deionized water droplets rebounding on the surface of F_16_CuPc-modified superhydrophobic film at a speed of 0.5 m/s. This resilient process contains four stages, which are falling, spreading, retraction, and rebound process. The falling process refers to the process before water droplets contact a solid surface, where water droplets mainly do free-fall under the action of gravity. The spreading process refers to the process by which the droplet expands to its maximum diameter from just touching the solid surface. The retraction process refers to the process from the maximum spreading state to leaving the solid surface. The rebound process refers to the process by which the droplet leaves from the solid surface to the highest position, which is the main process by which the kinetic energy of the droplet is converted into gravitational potential energy, and some smaller satellite droplets are generated if a strong adhesion exists between the droplet and the surface. As shown in Figure 5d, the water droplets show a complete rebound when dropping on the surface of the F_16_CuPc-modified biomimetic superhydrophobic film, indicating that there is subtle adhesion between the surface and the water droplets. These results demonstrate that the F_16_CuPc-modified biomimetic superhydrophobic film has excellent dynamic wettability, which can be attributed to the introduction of the F_16_CuPc modifier. The introduced F_16_CuPc film as a modifier resulted in a significant decrease in the surface energy, and decreased the adhesion force between the surface and water droplets.

In addition to the superior superhydrophobic properties, the stability of the superhydrophobic film also plays a critical role to realize the practical application. Therefore, the stabilities of the F_16_CuPc-modified biomimetic superhydrophobic film were investigated under different conditions. The changes of the WCA and WRA of the modified film under high-temperature (200 °C–400 °C) annealing for 1 h are depicted in Figure 6a. It can clearly be observed that the surface wettability of the sample hardly changed after heating at 200 °C for 1 h, and then slightly decreased with the further increase in the annealing temperature. Although the WRA increased from 0.5° to 7.2°, an eligible WCA of 159.6° was obtained after heating at 400 °C for 1 h. 

To reveal the origin of the reduced WRA and WCA with the increase in the annealing temperature, thermogravimetric analysis (TGA) was conducted for the F_16_CuPc-modified biomimetic superhydrophobic film. As shown in Figure 6b, the weight of the samples loses about 2% when the temperature is in the range of 200 °C to 400 °C, which mainly results from the loss of the adsorbed water in the molecular sieve and the decomposition of the −CH_3_ group in the PDMS into small-molecule gas [28]. With the further increase in the temperature, an obvious accelerated process was observed, in which the weight of the sample rapidly decreased. The weight loss increased to about 15% and 31% when the temperature reached 500 °C and 800 °C, due to the decomposition of the organic groups on the sample film. When the temperature reached 800 °C, a sluggish reduction was observed, indicating that the F_16_CuPc was almost completely decomposed and PDMS was also completely decomposed into SiO_2_. These results demonstrate that the F_16_CuPc-modified biomimetic superhydrophobic film still maintains superior superhydrophobicity under the influence of the high temperature of 400 °C, which could meet most of the needs in practical applications.

Mechanical robustness is a prerequisite for superhydrophobic films to be widely commercially applied. Figure 7 shows the changes of the WCA and WRA of the sample film after the wear test and tape stripping test. It can be observed that the WCA of the sample decreased from 167.1° to 157.3° after 30 wear tests under the action of a 500 N weight, and it shows strong water adhesion after 25 wear tests. Under the action of a 1000 N weight, the superhydrophobic properties of the surface decreased faster, the WCA decreased from 167.1° to 153.3°, and the sample surface showed strong water adhesion after 20 wear tests. For the tape stripping test, the WCA of the sample decreased from 167.1° to 131.5° after 20 tape stripping experiments, and it showed strong water adhesion after tests were conducted 15 times. The mechanical stability of the stacked substrate without the modification of F_16_CuPc was also investigated. As shown in Appendix A, the surface loses the superhydrophobic property after 20 wear tests with a 1000 N weight or 15 striping experiments. These results demonstrate that the F_16_CuPc-modified biomimetic superhydrophobic film has superior mechanical stability, compared to the stacked substrate without modifications.

The inferior UV stability always hampers the development of a superhydrophobic film based on the PDMS matrix. Under the irradiation of a UV lamp, the −OSi(CH_3_)_2_O− group on the surface of the PDMS would be transformed to −O_4_Si(OH)_4-n_ as well as −OH, −COOH, and −CO groups, and finally enhanced the hydrophilia of surface [41]. To study the UV stability of the F_16_CuPc-modified biomimetic superhydrophobic film, the changes in the sample surface wettability under the long-term irradiation via two different UV lamps are illustrated in Figure 8a,b. After 24 h of irradiation of 365 nm, the WCA of the sample surface decreased from 167.1° to 160.7°, while the WRA increased from 0.5° to 4.6°. The superhydrophobic performance of the surface decreased faster when the irradiation wavelength reduced to 254 nm, so that the WCA of the sample surface decreased from 167.1° to 157.1°, and the WRA increased to 10° after 24 h irradiation. For the stacked substrate without modification, the UV stability is summarized in Appendix A. The WCA reduced from 162.3° to 153.6° and 140.4° under the irradiation of 365 nm and 254 nm, respectively. The F_16_CuPc-modified substrate exhibited better UV light stability than the stacked substrate without modification. Appendix A shows the absorption spectrum of the pure F_16_CuPc and proves that F_16_CuPc has a favorable UV absorption capability, revealing that the modification of F_16_CuPc can reduce the penetration of the UV light into the film. The reduction in the UV penetration finally reduced the crosslinking and cracking reactions of the PDMS components and enhanced the UV stability of the F_16_CuPc-modified biomimetic superhydrophobic film.

Appendix A shows the change of the WCA and WRA of the F_16_CuPc-modified sample soaking under 20 cm water for 10 days. The hydrophobicity of the sample continuously decreased with the increasing soaking time. After soaking for 10 days, the WCA of the sample surface decreased from 167.1° to 135.2°, and the sample showed strong water adhesion after soaking for 6 days. The influence of water impact at different flow rates (2–10 m/s) on the surface wettability was also investigated. The superhydrophobicity of the sample slightly decreased with the increase in the water column velocity. The WCA and WRA of the sample surface remained at 157.6° and 7.2° under the 10 m/s water column for 1 min. The F_16_CuPc-modified biomimetic superhydrophobic film exhibited excellent stability due to the dense film formed by F_16_CuPc, which can play a certain role in buffering the impact of the water column, thus reducing the deformation of the surface hierarchical structure.

Finally, the chemical stability of the surface of the F_16_CuPc-modified biomimetic superhydrophobic film was measured. Appendix A shows the WCAs of the droplets with different pH values on the surfaces, with or without F_16_CuPc modification. For the F_16_CuPc-modified surface, the contact angle of the neutral droplet (deionized water), the acidic droplet with pH = 1, and the strongly alkaline droplet with pH = 13 were 167.1°, 150.3°, and 162.8°, respectively, while for the surface without modification, the contact angle reduced from 162.3° for the neutral droplet to 137.2° for the acidic droplet and 160.3° for the alkaline droplet, respectively. The results demonstrate that the F_16_CuPc-modified surface exhibits superior chemical stability. The stable benzene ring structure in the molecular guaranteed the stability of the F_16_CuPc and finally enhanced the stability of the F_16_CuPc-modified biomimetic superhydrophobic surface. The F_16_CuPc acted as a buffer layer and isolated the active metal oxides in the film when acid droplets fell on the surface, while an isolator was used to prevent the contact between the ZnO components on the film and the alkaline droplets. The F_16_CuPc-modified biomimetic superhydrophobic film has superior chemical stability, which enables the sample film to be used in harsh conditions. 

## 4. Conclusions

In summary, the F_16_CuPc was firstly used as the modifier to increase the WCA of the biomimetic micro-nanostructure, which was composed of PDMS, ZnO, and MS. With the introduction of F_16_CuPc, an enhanced superhydrophobic property was observed, where the WCA and WRA reached 167.1° and 0.5°, respectively. The superhydrophobic surface had a biomimetic micro-nanostructure and kept the water droplets on the surface in the Cassie state, presenting an excellent self-cleaning capability with a simple fabrication process. In addition, SEM, EDS, and XPS analyses were carried out and the results verify that the enhanced superhydrophobic properties of the film are attributed to the modification of the F_16_CuPc. Finally, the stability of the F_16_CuPc-modified biomimetic superhydrophobic film was investigated through a series of experiments. The thermal stability, mechanical stability, and chemical stability of the superhydrophobic surface and the influence of UV radiation, underwater immersion, and water impact on the superhydrophobic properties were studied. The results confirm that the sample film had excellent stability and environmental adaptability. These superior stabilities were mainly ascribed to the high thermal stability and UV absorption capacity of F_16_CuPc, which acted as a dense protective layer to reduce the attenuation of surface superhydrophobic properties. This biomimetic micro-nanostructure superhydrophobic fabrication process may be a potential direction for superhydrophobic fabrication, due to its simple fabrication process, excellent superhydrophobic property, and favorable stability.

## Figures and Tables

**Figure 1 nanomaterials-12-00953-f001:**
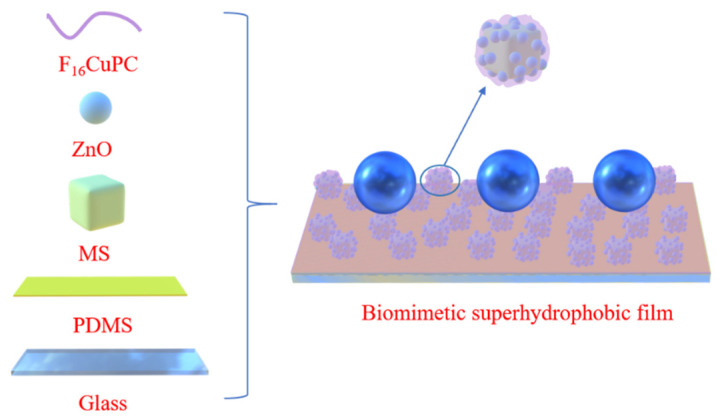
Schematic diagram of the biomimetic superhydrophobic film.

**Figure 2 nanomaterials-12-00953-f002:**
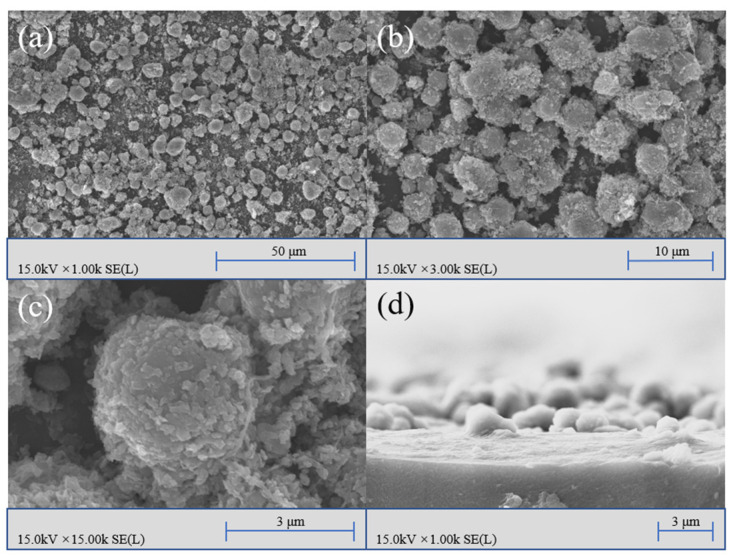
SEM images of the biomimetic superhydrophobic film surface modified by F_16_CuPc with different magnifying factors, (**a**) 1 k, (**b**) 3 k, and (**c**) 15 k, and (**d**) the cross-section image of the F_16_CuPc-modified biomimetic superhydrophobic film.

**Figure 3 nanomaterials-12-00953-f003:**
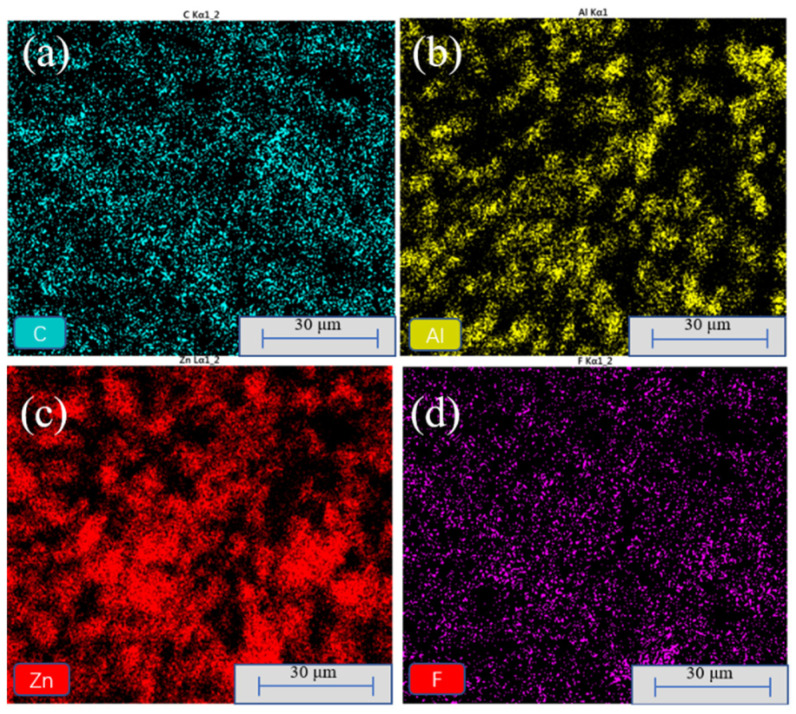
EDS analysis of the biomimetic superhydrophobic films modified by F_16_CuPc (**a**) C, (**b**) Al, (**c**) Zn, and (**d**) F elements.

**Figure 4 nanomaterials-12-00953-f004:**
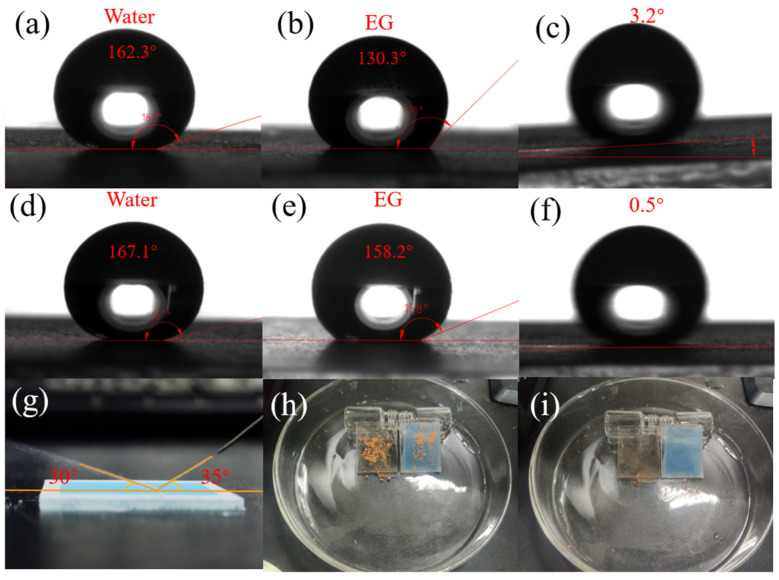
The WCA, EGCA, and WRA on (**a**–**c**) the stacked substrate and (**d**–**f**) the F_16_CuPc-modified biomimetic superhydrophobic surface. (**g**) The water column reflects images on the F_16_CuPc-modified biomimetic superhydrophobic surface. Images (**h**) before and (**i**) after the self-cleaning of the glass (left) and the F_16_CuPc-modified biomimetic superhydrophobic surface (right).

**Figure 5 nanomaterials-12-00953-f005:**
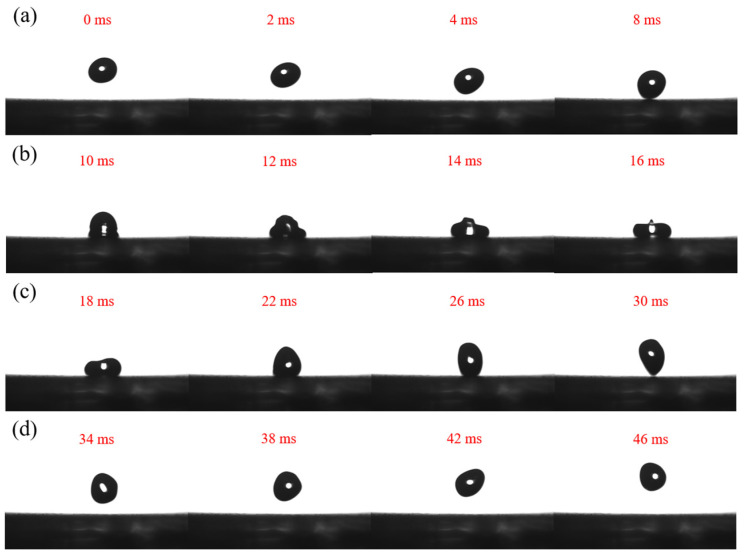
Images of the water droplets striking the biomimetic superhydrophobic films modified by F_16_CuPc, (**a**) the falling process, (**b**) the spreading process, (**c**) the retraction process, and (**d**) the rebound process.

**Figure 6 nanomaterials-12-00953-f006:**
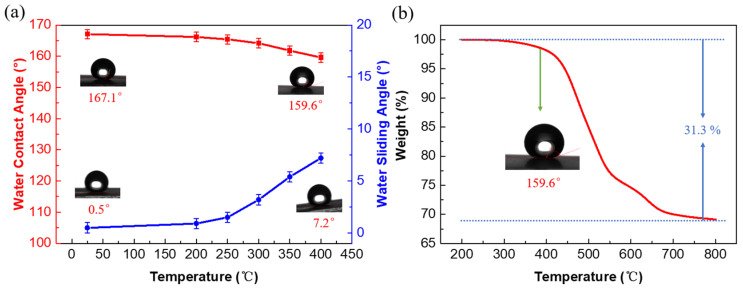
(**a**) The changes of the WCA and WRA of the modified film under heating for 1 h; (**b**) the thermogravimetric analysis of the modified film.

**Figure 7 nanomaterials-12-00953-f007:**
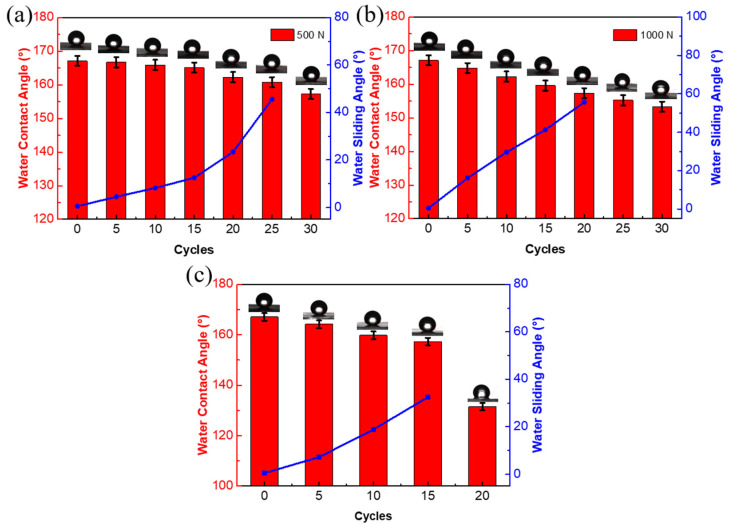
The changes of the WCA and WRA of the F_16_CuPc-modified film after the wear test with (**a**) 500, (**b**) 1000 N weight, and (**c**) tape stripping experiment.

**Figure 8 nanomaterials-12-00953-f008:**
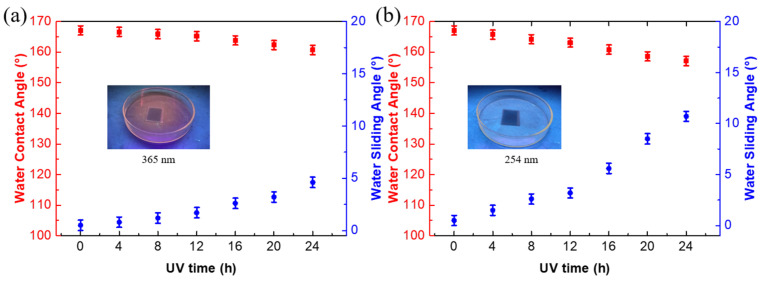
The changes of the WCA and WRA of the modified film after UV irradiation with a (**a**) 365 nm and (**b**) 254 nm UV lamp.

## Data Availability

Not applicable.

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
