# Peer review of "Biomimetic Superhydrophobic Films with an Extremely Low Roll-Off Angle Modified by F16CuPc via Two-Step Fabrication"

_nanomaterials, 2022, doi:10.3390/nano12060953_

Round 1

Reviewer 1 Report

In going over the manuscript, the novelty and the importance of the results is not clear. For example, the authors write - "This biomimetic micro-nanostructure super- hydrophobic fabrication process may be potential direction for superhydrophobic fabrication in daily life due to its simple fabrication process, excellent superhydrophobic property, and favourable stability.” It would help if the authors precisely describe the importance of the study. What do the authors mean by "fabrication in daily life"?

The manuscript reads like a technical report with a very weak emphasis on the Results and Discussion component of the manuscript.  It would help if the authors can compare their results with those in the literature.

What is the importance of Figure 3?  How does the EDS analyses relate to the study?

In general, the manuscript needs to be improved significantly by addressing the following:

Novelty of the Study;

Importance of the Study;

Application of the Study?

Comparison with the Literature - How are the results in this study comparable to those in the literature? Do they agree? Or are they different from those in the literature?

What is the influence of surface conditions in Figure 5?

What is the influence of temperature?

The language in the manuscript can be improved throughout; more importantly, authors need to be very clear about their thoughts and interpretation.

Reviewer 2 Report

Comments to “Biomimetic superhydrophobic films ………” Hu et al.

Procedures are not well described. Quantitative comparison with other systems are not reported. Quality of figures is poor; characters in figures and axes must be enlarged in order to be read clearly. Comparisons of the behaviour with and without F16CuPc are not reported.

English need major revision.

Introduction section contains results and conclusions, which is not adequate.

Experimental section: MS is not described nor explained how it is obtained.

“annealed for a period of time”, it is not precise. Indicate at least a time interval.

How F16CuPc was evaporated? Spontaneously or by heating?

Which are the specifications of the used water? (Quality or purity, pH)

EG is not indicated here, and its specifications.

Wear and tape stripping tests are not described here.

Where F16CuPc film is in Figure 1?

 Results: the meaning of EGCA is not described.

Calculation of surface energy is not reported adequately.

Ln 120 “the 50 nm F16CuPc film”. What it means?

Ln 127-129 are confusing, “the two-level structures … before and after …”!!

Some peaks are not identified in figure S3.

Figure 4: I do not understand the caption figure for images e) and f). Some particular symbols appear in a) and b) images.

Figures 4-8: numbers are difficult to read.

Figure 6b is cited before Figure 6a.

Different notation is used along the text: “biomimetic” or “bionic” film.

Authors must explain how contact angle was measured at high temperatures, even above 100ºC.

Ln 226-228: “These results demonstrated that the F16CuPc modified film had more stable mechanical stability than the film without modification”, but the results of the film without modification are not shown, and neither are shown for another comparisons.

Conclusions and abstract are very similar.

Supplementary: What means pure F16CuPc or pure F16CuPc film versus F16CuPc modified film? Revise notation, because I do not understand so many notations.

In Fig S2-S4, the 16 of F16 must be sub-index.

Figures S5, S6: captions are not self-explaining. Which system is there presented?

Fig S5: a) and b) are not indicated in the figure.

Round 2

Reviewer 1 Report

The revised manuscript is recommended for publication in Nanomaterials.

However, the language in the manuscript is highly incorrect and requires significant edits and corrections.

It is extremely difficult to comprehend the authors' writing.  Readers will not understand the interpretation of the authors.

Author Response

Thanks very much for the reviewer’ recommendation and suggestion.

For the English, we tried our best to improve the grammar and readability of the revised manuscript. Changes are highlighted with “Track change” in the revised manuscript. We hope these revisions could be sufficient to have the paper accepted for publication.

Reviewer 2 Report

The paper has been improved but some points remain.

English still needs minor revision.

Section 2.2. Ln 75: “The F16CuPc of 50 nm”, thickness is missed.

Section 3. ln 231: “The results demonstrated that the F16CuPc-modified surface exhibits superior chemical stability” and ln 235 “The F16CuPc-modified biomimetic superhydrophobic film has superior chemical stability”, but this has not been proved in the paper since results of the un-modified film were not reported.
